# Real Time Analysis of Bovine Viral Diarrhea Virus (BVDV) Infection and Its Dependence on Bovine CD46

**DOI:** 10.3390/v12010116

**Published:** 2020-01-17

**Authors:** Christiane Riedel, Hann-Wei Chen, Ursula Reichart, Benjamin Lamp, Vibor Laketa, Till Rümenapf

**Affiliations:** 1Institute of Virology, Vetmeduni Vienna, 1210 Vienna, Austria; Hann-Wei.Chen@vetmeduni.ac.at (H.-W.C.); Till.Ruemenapf@vetmeduni.ac.at (T.R.); 2VetCore Facility for Research, Vetmeduni Vienna, 1210 Vienna, Austria; ursula.reichart@vetmeduni.ac.at; 3Institute of Virology, Faculty of Veterinary Medicine, Justus-Liebig University, 35392 Gießen, Germany; Benjamin.J.Lamp@vetmed.uni-giessen.de; 4Department of Infectious Diseases, Virology, University of Heidelberg, 69120 Heidelberg, Germany; vibor.laketa@med.uni-heidelberg.de; 5German Center for Infection Research, 69120 Heidelberg, Germany

**Keywords:** *Pestivirus*, BVDV, CD46, life cell imaging, attachment, surface transport

## Abstract

Virus attachment and entry is a complex interplay of viral and cellular interaction partners. Employing bovine viral diarrhea virus (BVDV) encoding an mCherry-E2 fusion protein (BVDV_E2-mCherry_), being the first genetically labelled member of the family *Flaviviridae* applicable for the analysis of virus particles, the early events of infection—attachment, particle surface transport, and endocytosis—were monitored to better understand the mechanisms underlying virus entry and their dependence on the virus receptor, bovine CD46. The analysis of 801 tracks on the surface of SK6 cells inducibly expressing fluorophore labelled bovine CD46 (CD46_fluo_) demonstrated the presence of directed, diffusive, and confined motion. 26 entry events could be identified, with the majority being associated with a CD46_fluo_ positive structure during endocytosis and occurring more than 20 min after virus addition. Deletion of the CD46_fluo_ E2 binding domain (CD46_fluo_∆E2bind) did not affect the types of motions observed on the cell surface but resulted in a decreased number of observable entry events (2 out of 1081 tracks). Mean squared displacement analysis revealed a significantly increased velocity of particle transport for directed motions on CD46_fluo_∆E2bind expressing cells in comparison to CD46_fluo_. These results indicate that the presence of bovine CD46 is only affecting the speed of directed transport, but otherwise not influencing BVDV cell surface motility. Instead, bovine CD46 seems to be an important factor during uptake, suggesting the presence of additional cellular proteins interacting with the virus which are able to support its transport on the virus surface.

## 1. Introduction

Before entering a host cell, viruses have to establish contact with the cell and travel to sites that are suitable for entry. This process of attachment can involve different receptor molecules and transport mechanisms, which can be classified into diffusion, drifts, and confinement [1]. Diffusive motion can be caused by interaction of viruses with a number of receptor molecules that is too low to cause confinement or can support the screening of the cell surface for suitable sites of endocytosis. Directed motion is the result of interaction of the virus with cellular proteins or lipid structures that are linked to F-actin. Understanding the extracellular movement of virus particles, as well as the kinetics of entry and protein expression, provides important insights into the key players involved in attachment and entry and in infection dynamics in general.

For the family *Retroviridae*—facilitated by the ease of generation of genetically labelled virus particles—attachment and entry dynamics have been studied excessively. Virus surfing—the directed, actin-dependent transport of virus particles on the outside of filopodia, cytonemes, and retraction fibres—was also discovered employing a member of the *Retroviridae*, namely murine leukemia virus (MLV) [2]. This mode of extracellular, directed transport has since been described for members of the *Adenoviridae* [1], *Herpesviridae* [3], and *Papillomaviridae* [4].

For the medically relevant family *Flaviviridae*, the lack of genetically labelled virus particles has been overcome by the utilization of lipophilic dyes or covalently linked fluorophores [5,6,7,8]. These studies demonstrated the diffusive movement of dengue virus along the cell surface to clathrin coated pits and the subsequent association with specific markers of endocytosis [8] and helped in the identification of T cell immunoglobulin mucin-1 as a dengue virus receptor [9]. Also, labelled hepatitis C virus (HCV) particles demonstrated the involvement of actin in cell surface and intracellular virus transport [7].

The genus *Pestivirus* is also part of the *Flaviviridae* and characterized by the presence of three viral surface glycoproteins—E^rns^, E1, and E2—and an additional N-terminal protease, N^pro^. Members of this genus are pathogens of cloven-hoofed animals, including the highly economically relevant pathogens bovine viral diarrhea virus (BVDV) and classical swine fever virus (CSFV). In recent years, genetically labelled BVDV and CSFV clones have been constructed based on N-terminal fusion of luciferase or fluorophores to either E^rns^ or E2 [10,11,12,13]. For a BVDV E2-fluorophore fusion protein, visualization of purified particles in fluorescence microscopy based on the specific fluorescence signal has been reported [13]. BVDV enters host cells after initial interactions of E^rns^ with heparan sulphates [14] and subsequent binding of E2 to its cellular receptor, bovine CD46 [15,16], by clathrin mediated endocytosis [17,18] and fusion occurs after endosomal acidification [18,19]. Interestingly, recent evidence implies that BVDV is preferably transmitted by direct cell-to-cell spread in a CD46 independent manner [12], indicating the involvement of additional factors in BVDV spread. The BVDV-1 clone employed in [12] also encodes for an mCherry-E2 fusion protein, in the backbone of strain NADL, demonstrating the utilization of different transmission modes in the presence of an E2-fusion protein.

Bovine CD46 is an ubiquitously expressed, type I transmembrane glycoprotein and exists as different splice variants, affecting the length of its heavily O-glycosylated, membrane proximal regions (STP) and its cytoplasmic C-terminus. The membrane distal, extracellular part of the protein consists of four complement control protein modules (CCP) which have been implicated in the binding of a variety of pathogens, leading to its description as a “pathogens magnet” [20]. Physiologically, CD46 is a cofactor of the inactivation of complement components C3b and C4b and also involved in T cell regulation, modulation of autophagy and reproductive biology (reviewed in [21]). Its surface levels are regulated by clathrin-dependent endocytosis [21] or micropinocytosis after cross-linking [22]. BVDV E2 binding to CD46 is mediated by the 30 C-terminal amino acids of CCP-1 [16]. Interestingly, CD46’s physiological ligands mainly interact with the membrane proximal CCPs 3–4, while pathogens mostly interact with the membrane distal CCPs 1–2 [23].

While the function of bovine CD46 as a receptor for BVDV is well documented in the literature, it is currently unclear what its exact functions are during virus attachment and entry. Also, it has been shown that bovine CD46 is not sufficient for entry. To determine which stages of the entry process—attachment, surface motion, or uptake—are affected by CD46, we conducted life cell imaging experiments employing purified, fluorophore-E2 labelled BVDV particles and SK6 cell lines inducibly expressing fluorophore labelled CD46 with and without the virus interacting CCP-1 module [16].

## 2. Materials and Methods

### 2.1. Viruses and Cells

For propagation, cells were cultured in DMEM (Capricorn, Ebsdorfergrund, Germany) supplemented with 10% FCS (Bio & Sell, Feucht/Nürnberg, Germany) and penicillin/streptomycin (Merck, Darmstadt, Germany) at 37 °C, 5% CO_2_. SK6 tet-on cells inducibly expressing bovine CD46 labelled with mCherry or mClover (CD46_fluo_) have been described in [13]. The E2 binding CCP of CD46—as reported by [16]—was deleted by PCR (Vazyme Biotech, Nanjing, China) in the previously described CD46_fluo_-encoding construct employing the following primers (Eurofins, Ebersberg, Germany): forward: cgaCGGTGTCCTACCCTAGCTGATC; reverse: GGCATCGGAGGACGTGGGCAG, resulting in CD46_fluo_∆E2bind. SK6 tet-on cells were transfected with CD46_fluo_∆E2bind by electroporation and clonally selected as described in [13].

For the determination of susceptibility of SK6 CD46_fluo_ or CD46_fluo_∆E2bind cells to BVDV_E2-mCherry_, 2 × 10^5^ MDBK, SK6 CD46_fluo_ or CD46_fluo_∆E2bind cells or SK6 cells inducibly expressing GFP were seeded in each well of a 24-well plate 24 h before the start of the experiment. SK6 cells are of porcine origin and porcine CD46 is not a receptor for BVDV [16]. Cells were induced 16 h before infection with 2.5 µg/mL doxycycline (Merck, Darmstadt, Germany). Cells were infected with ten-fold dilutions of a BVDV-1 strain expressing mCherry fused to the N-terminus of E2 (BVDV_E2-mCherry_) [13] for 4h. Thereafter, medium was exchanged to medium containing 1% carboxymethylcellulose to prevent virus transfer via the culture medium. 48 h after infection, infected foci were quantified by fluorescence microscopy (Olympus IX-70, Tokyo, Japan) to determine the titer and the susceptibility was calculated in % by dividing the titer determined on a given cell line through the titer determined on MDBK cells. BVDV_E2-mCherry_ was propagated on MDBK cells in 5-layer tissue culture flasks (Corning, Corning, NY, USA) and purified and concentrated as described in [13]. The sequence of the plasmid encoding the full sequence of BVDV_E2-mCherry_ is provided as Appendix A.

For life cell imaging, 2 × 10^4^ SK6 CD46_fluo_ or CD46_fluo_∆E2bind cells were seeded in each well of an IBIDI µ slide 8 well chamber slide 24 h before the start of the experiment in medium devoid of phenol red. 16 h before the start of the experiment, expression of CD46_fluo_ or CD46_fluo_∆E2bind was induced by addition of 2.5 µg/mL doxycycline (Merck, Darmstadt, Germany).

### 2.2. Life Cell Imaging

All data was acquired on an Andor Revolution spinning disk confocal microscope (Oxford Instruments, Abingdon, UK) based on a Yokogawa CSU W1 with Nikon Ti2 stand equipped with two Andor DU-888 cameras. Imaging was performed in a humidified chamber at 37 °C and 5% CO_2_. An APO TIRF NA 1.49 100× magnification oil immersion objective was used, resulting in a lateral pixel size of 0.13 µm. For the detection of CD46_fluo_ and CD46_fluo_∆E2bind, mClover was excited by a 515 nm laser for 0.1 s and the emitted signal was bandpass filtered with a 540/30 nm filter. E2-mCherry was excited by a 594 nm laser for 0.1 s and the emitted light was bandpass filtered with a 647/57 nm filter before detection. Excitation and detection of each fluorophore was performed consecutively.

For the examination of virus attachment, cell surface transport, and entry, one frame/10 s was recorded starting directly, 5, 10, 15, 20, or 25 min after virus addition (multiplicity of infection (MOI) = 10) and data was acquired for 5 or 10 min. Three focal planes (*Z*-step size = 0.8 µm) were acquired for each time point to compensate for potential focusing errors and to increase the area accessible for analysis. The *z*-level was chosen close to the cover slip to allow imaging of filopodia, retraction fibers, lamellipodia, and the lamella. In order to judge intracellular background levels during E2-mCherry excitation, one *z*-stack was acquired of each field of view to be employed in the experiment before addition of virus.

To follow the time course of infection, cells were imaged at one *z*-level every 10 min starting 60 min after infection until 16 h after infection. Two different MOIs (1 and 10) were employed to assess the effect of different MOIs on the development of E2-mCherry signal. Due to phototoxicity, it was not possible to acquire several *z*-levels.

### 2.3. Image Processing and Analysis

All image processing and particle tracking was performed in ImageJ [24]. Raw frames were filtered with a Gaussian filter (sigma = 1.4) to improve data visualization. For initial particle tracking, maximum intensity projections along the *z*-axis were generated and particles tracked employing ImageJ’s [24] manual tracking plug-in. Subsequently, tracks were verified on the full z-stack and their localization (lamellipodium, lamella, cell body) as well as their direction of movement (towards, tangential, away, random, with reference to the cell body) and the association with CD46_fluo_ or CD46_fluo_∆E2bind was documented. A python script (Appendix A) was used to analyze the output of the manual tracking plug-in with regard to direct and relative distance covered by a given particle, directionality (ratio direct to relative distance), average, minimum and maximum speed as well as the standard deviation of speed for a given track. The mean squared displacement (MSD) was calculated in R employing a customized script based on the code developed by [25]. Fits of the MSD curves were employed to calculate the velocity (V) of directed movement in case of an exponential slope using the following formula: $y=V^{2}x^{2}+D_{0}x+a$ and the diffusion coefficient (D_0_) was calculated using $y=4D_{0}x+a$ in case of a linear slope. A two-tailed Student’s *t*-test was performed to assess the likelihood of significant differences.

## 3. Results

### 3.1. BVDV Entry Is a Slow Process

To assess the infection cycle of BVDV in real time, we employed a previously described labelling strategy of the viral E2 surface glycoprotein [12,13]. SK6 cells inducibly expressing the fluorophore labelled, cellular surface receptor, bovine CD46 were chosen as system to study the infection cycle, as these cells demonstrated a high susceptibility to infection with BVDV after induction. SK6 cells not expressing bovine CD46 display a low susceptibility to BVDV infection, which is not mediated by porcine CD46, as porcine CD46 has previously been shown not to be involved in BVDV invasion [16]. After initial experiments employing either a system of mClover labelled BVDV and mCherry labelled CD46 or vice versa, we decided to employ mCherry labelled BVDV (BVDV_E2-mCherry_) and mClover labelled CD46 (CD46_fluo_), as strong photobleaching of mClover labelled BVDV rendered it inapplicable for life cell imaging.

Time series of 5 or 10 min duration with a frame rate of one frame/10 s were acquired at different time points (0–25 min in 5 min steps) after addition of BVDV_E2-mCherry_ (MOI 10). Association of mCherry positive particles with the surface of SK6 CD46_fluo_ cells could readily be observed and 801 particles were tracked by hand (*n* cells = 160). Due to the low signal to noise ratio, automatic tracking approaches did not improve the ease of analysis and were therefore not utilized. Entry events were rarely observed (*n* = 26, 2.8% of total particles tracked) and the majority occurred more than 20 min after virus addition (Figure 1A). In 65.5% of these entry events, the signal of the virus was associated with a CD46_fluo_ positive vesicular structure. Association with such a vesicle was maintained after endocytosis in 76% of events. After endocytosis, particles could on average be tracked for 2.8 min and migrated on average 3 µm inside the cell (Figure 1A). Two examples of mCherry positive particles entering a cell are shown in Figure 1B,C and Appendix A.

Due to the intracellular background level in the mCherry emission range, the amount of intracellular, mCherry positive foci and their migration behavior was not analyzed as an unambiguous differentiation between background and specific signal was not possible.

Particles travelled with an average speed of 0.062 µm/s (s.d. 0.036 µm/s), an average maximum speed of 0.147 µm/s (s.d. 0.15) and an average directionality—defined as the ratio of direct distance versus real distance covered by a particle—of 0.34 (s.d. 0.22) on SK6 CD46_fluo_ cells.

Directed transport of viruses along the outside of filopodia, cytonemes, and retraction fibers has been described for enveloped and non-enveloped viruses. The potential usurpation of this transport route was hence also examined in the context of BVDV_E2-mCherry_. Virus surfing could be observed for 16 particles (2.0% of total tracks, Appendix A). These particles migrated with an average velocity of 0.070 µm/s and an average directionality of 0.446, indicating an advantage of this type of transport regarding movement in a given direction. The average maximum speed of these particles was 0.176 µm/s, indicating a potential coupling to retrograde actin flow [2,26].

### 3.2. First Detection of BVDV E2-mCherry Signal Is Depending on MOI

In order to further analyze the progression of infection and to visualize the intracellular distribution of E2-mCherry over time, time series starting 60 min after virus addition and running for 16 h (1 frame every 10 min) were recorded. In all cells observed, an evenly distributed, slightly granular E2-mCherry signal was present from the initial detection of a specific signal (Appendix A). This intracellular distribution pattern was reminiscent of an ER staining pattern, which is in good accordance with the localization of E2 in the ER lumen as already reported by [27,28]. Interestingly, the time after which an E2-mCherry signal could be resolved correlated with the MOI employed in a given experiment. For an MOI of 1, E2-mCherry could be detected on average 645 min after virus addition (*n* = 13) (Figure 2A, Appendix A), whereas E2-mCherry could already be resolved on average 195 min (*n* = 16, Appendix A) after infection if an MOI of 10 was used.

To gain further insights into the dynamics of E2-mCherry trafficking 20 h after infection, cells were imaged for 5 min with a frame rate of 1 frame/10 s. In addition to the diffuse, granular staining pattern of E2-mCherry, point-shaped, high signal intensities could be observed (Figure 2B, Appendix A). They were partially associated with high CD46_fluo_ signal intensities and this association was maintained during the whole course of the experiment. High intensity E2-mCherry foci could both be stationary or highly mobile (up to 0.5 µm/s), especially in the cell periphery (Appendix A).

### 3.3. CD46_fluo_ Decreases the Speed of Directed Surface Motion of BVDV_E2-mCherry_

In order to elucidate the effect of specific receptor binding on virus entry, SK6 cells inducibly expressing fluorophore-labelled CD46 with a deleted E2-binding domain (CD46_fluo_∆E2bind) were generated based on previous results by [16], reporting the CCP-1 as interaction partner. These cells showed a susceptibility to BVDV comparable to the susceptibility of SK6 cells inducibly expressing GFP (Figure 3). GFP expressing cells were chosen as control to account for a potential effect of induced expression on virus replication. The susceptibility of SK6 CD46_fluo_∆E2bind cells was more than 250-fold reduced in comparison to SK6 cells expressing unmodified CD46_fluo_ after induction of expression.

This system was chosen over experiments employing inhibition of virus attachment by antibodies targeting neutralizing epitopes of E2 or functionally important domains of CD46 as it—in our opinion—provided the most direct and defined system to study the role of E2-CD46 interaction in the viral life cycle by life cell imaging. An inhibition of infection by BVDV tagged with a fluorophore at the E2 N-terminus employing before mentioned antibodies has already been demonstrated by [12] and this inhibition is comparable to BVDVs not bearing a tag at the E2 N-terminus. This clearly demonstrates the importance of the same domains in CD46-E2 interactions.

To assess the effect of receptor binding on the surface movement of BVDV_E2-mCherry_, 1081 mCherry-positive particles were tracked on SK6 CD46_fluo_∆E2bind cells (*n* cells = 297). Of those 1081 particles, only 2 particles (0.2%) could be observed entering a cell, which is 7% of the entry events found for SK6 CD46_fluo_ cells. One of these events was associated with a CD46_fluo_∆E2bind after uptake.

In order to better understand the movement of particles on the surface of SK6 CD46_fluo_ and SK6 CD46_fluo_∆E2bind cells, particle trajectories were further analyzed. The average direct distance covered on the cell surface during a given observation period was 3.15 µm (s.d. 2.71 µm) for SK6 CD46_fluo_ and 2.83 µm (s.d. 2.18 µm) for SK6 CD46_fluo_∆E2bind cells (Figure 4), while the average real distance covered was 10.9 µm (s.d. 8.64 µm) for CD46_bov_ and 9.14 µm (s.d. 6.06 µm) for CD46_bov_∆E2bind. The average directionality—defined as the quotient of direct versus real distance—was 0.34 (s.d. 0.22) for SK6 CD46_bov_ and 0.37 (s.d. 0.23) for SK6 CD46_bov_∆E2bind cells and therefore seems unaffected by CD46 binding. Particles on SK6 CD46_fluo_ cells travelled with an average velocity of 0.062 µm/s (s.d. 0.036 µm/s) and an average maximum speed of 0.147 µm/s (s.d. 0.15). Particles on SK6 CD46_fluo_∆E2bind cells were characterized by an average velocity of 0.063 µm/s (s.d. 0.028 µm/s) and an average maximum speed of 0.136 µm/s (s.d. 0.063 µm/s), indicating no effect of CD46 binding on observed velocities.

On both, SK6 CD46_fluo_ and SK6 CD46_fluo_∆E2bind cells, the majority of particles was localized on the lamella or lamellipodium up to 10 min after infection; beginning with 15 min after infection, the majority of particles was bound to the cell body (Appendix A). On average 51% of the particles on SK6 CD46_fluo_ cells were associated with an increased CD46 signal intensity for at least 3 consecutive frames. On SK6 CD46_fluo_∆E2bind cells, this percentage increased to an average of 74%. To assess potential differences in particle travelling directions, the overall direction of movement of a given particle was described as moving random or away from, tangential to or towards the cell body. Such a direction could unambiguously be assigned to 423 tracks on SK6 CD46_fluo_ cells and 705 tracks on SK6 CD46_fluo_∆E2bind cells (Appendix A). 3% of particles tracked on SK6 CD46_fluo_ cells and 12% of particles tracked on SK6 CD46_fluo_∆E2bind cells showed a movement away from the cell body, whereas 57% and 42% respectively exhibited random movement. Tangential movement was observed for 17% of particles on SK6 CD46_fluo_ cells and 18% of SK6 CD46_fluo_∆E2bind cells. 23% and 29% of particles exhibited movement towards the cell body on SK6 CD46_fluo_ cells and SK6 CD46_fluo_∆E2bind cells, respectively. These results indicate that the major difference in the direction of particle transport is a four times reduced number of particles moving away from the cell body if CD46 is functional.

To further characterize the number of tracks following a certain type of motion on the cell surface and to determine transport velocities and the diffusion coefficients (D_0_), MSD analysis was performed. Within both datasets, tracks exhibiting characteristics of directed motion, as well as limited diffusion or purely diffusive motility, could be identified. For SK6 CD46_fluo_ cells, 86 tracks (10.7% of total tracks, 47.5% of tracks in MSD analysis) exhibited linear slopes, indicative of purely diffusive motion, while 46 tracks (4.3% of total tracks, 31.7% of tracks in MSD analysis) were identified for SK6 CD46_fluo_∆E2bind cells (Figure 5A). The calculated diffusion coefficients were 0.020 µm^2^/s (s.d. 0.017 µm^2^/s) for SK6 CD46_bov_ cells and 0.015 µm^2^/s (s.d. 0.019 µm^2^/s) for SK6 CD46_fluo_∆E2bind cells and were not significantly different (*p* = 0.103). 70 tracks exhibiting an exponential slope could be identified for SK6 CD46_bov_ cells (8.7% of total tracks, 38.7% of tracks in MSD analysis) and 61 for SK6 CD46_bov_∆E2bind cells (5.6% of total tracks, 42.1% of tracks in MSD analysis) (Figure 5B). The calculated average particle velocity was 0.028 µm/s (s.d. 0.014 µm/s) for SK6 CD46_fluo_ cells and 0.039 µm/s (s.d. 0.018 µm/s) for SK6 CD46_fluo_∆E2bind cells, suggesting a significantly increased directional transport speed on SK6 CD46_fluo_∆E2bind cells (*p* < 0.001). Twenty-five tracks on SK6 CD46_fluo_ cells (3.1% of total tracks, 13.8% of tracks in MSD analysis) and 38 tracks on SK6 CD46_fluo_∆E2bind cells (3.5% of total tracks, 26.2% of tracks in MSD analysis) showed characteristics indicative of limited diffusion in MSD analysis.

## 4. Discussion

Virus attachment and entry are governed by various affinities and highly dynamic processes. For BVDV, our findings suggest the presence of different types of motility on the cell surface, being diffusion, directed motion, and confined diffusion. This is different from observations of Dengue virus, as the main motion type until the association with a clathrin-coated pit is diffusion [8]. Virus surfing, as a movement frequently utilized by other viruses (e.g., MLV), was rarely observed, albeit the presence of filopodia, cytonemes, and retraction fibers, implying a minor role of this transport mechanism for BVDV, at least in this experimental setup. All motion types could be observed independent of the presence or absence of the BVDV-binding competent cellular receptor, CD46 and track properties were comparable. Similarly, the proportions of cellular compartments—lamellipodium, lamella, and cell body—with associated virus particles were not affected by CD46’s ability to bind E2 when comparing different time points after infection. However, when analyzing directed motion in more detail by MSD analysis, track velocities were significantly increased if CD46 was not able to interact with E2. This might suggest that another interaction partner on the cellular surface—for example heparan sulphate or an as of yet unidentified attachment factor—might be responsible for coupling to the underlying actin network and that CD46 is rather involved in deceleration of particle transport on the cell surface or the anchoring of particles at potential sites of endocytosis. The importance of attachment factors such as heparan sulphate has also been demonstrated for other members of the *Flaviviridae* (reviewed in [29]). Also, CD46 is not sufficient to render cell lines derived from non-cloven-hoofed animals or humans susceptible to infection with BVDV. Instead, it increases the amount of cell-associated virus [15]. This indicates that additional cellular factors are required for successful virus infection after binding to CD46. Interestingly, overexpression of CD46_fluo_∆E2bind resulted in a further drop of susceptibility to BVDV, which might indicate the sequestration of a potential additional entry factor by CD46_fluo_∆E2bind.

In general, the number of observable entry events into SK6 CD46_fluo_ was very low (2.8% of tracked particles). This might indicate that cellular uptake of BVDV is a rather inefficient process. However, during data acquisition, we were not able to sample the whole cellular surface due to the short frame interval and phototoxicity. The focus of acquisition was on the cellular surface in contact with the growth support, to clearly depict cellular protrusions, the lamellipodium and the lamella. Particles could frequently be observed exiting the acquisition planes to regions of the cell body that were inaccessible in the chosen acquisition scheme. It is therefore possible that the low amount of observed entry events is—at least partially—due to a preference of BVDV to enter cells at specific areas of the cell body. Also, the specific infectivity of BVDV_E2-mCherry_ is below 1:10, meaning that more than 10 genome equivalents are needed for successful establishment of infection. It is currently unclear what particle defects are responsible for this phenomenon, but it has to be taken into consideration that a certain number of particles are already unable to evoke cellular uptake. In order to clarify these open questions and to also examine the entry process of BVDV into primary host cells, MDBK CD46 knock-out cell lines would be a valuable tool for future elucidation of the entry process.

Contrasting the increased speed of particle transport, cell entry events were very rare if CD46 was E2-binding incompetent. This, in combination with the rather rare CD46—virus co-migration on the cell surface, might implicate that CD46 is important for the signaling to evoke endocytosis, but not for surface movements towards potential sites of endocytosis. The frequently observed association of virus particles with CD46 positive vesicles after endocytosis and their comigration for several frames further supports this hypothesis, but additional experiments will be required for verification. Still, BVDV_E2-mCherry_ is able to enter host cells independent of a functional CD46 molecule. This strongly indicates the potential to use other entry receptors, but with much lower efficiency. The importance of factors different from CD46 in BVDV propagation is also highlighted by recent results reporting that BVDV cell-to-cell spread is independent of CD46 [12]. 

The CD46 splice variant employed in this study is identical to the bovine CD46 variant initially reported by [15]. Unfortunately, we were not able to elucidate the effects of different allelic versions of CCP-1 and different CD46 splice variants on the dynamics of BVDV entry due to the huge efforts of data analysis. A different permissiveness to infection of these different allelic and splice variants has however already been demonstrated [30] and their effect on the dynamics of BVDV entry will be an interesting topic of future studies.

Taking advantage of the easy tracking of virus protein production in infected cells, the effect of different MOIs on the earliest time point of specific fluorescence detection was examined. The highly significant difference observed leads to the conclusion that signal development is depending on the virus dose applied and that superinfection exclusion is not acting fast enough to prevent the infection of one cell with several particles in the context of this experimental setup [31]. Surveillance of the signal development of BVDV_E2-mCherry_ after infection revealed an initially finely granular, likely ER-associated [27,28] distribution throughout the whole cell, which intensified over time and finally also developed strong signal foci. Therefore, it seems likely that E2 is quickly distributed throughout the cell and not restricted to and radially expanding from a potential initial replication site. To better understand viral replication sites, additional studies are warranted to elucidate how the intracellular distribution of E2 is reflecting the overall distribution of non-structural proteins like NS3 and NS5B as part of the replication complex. This, in combination with ultrastructural studies, will provide further insights into the morphology of replication sites and allow comparison to the detailed morphological studies already performed on the membranous replication compartments of HCV [32] and flaviviruses [33,34]. The occasional association of CD46 positive intracellular foci with E2 mCherry positive foci suggests that CD46 is usually excluded from E2 mCherry containing compartments or only present in amounts below the detection limit of this experimental setup. Whether this association is unfavorable for the virus or serves an as of yet unknown purpose—like facilitation of exocytosis by taking advantage of CD46 transport to the cell surface—will need to be elucidated in the future.

In the presented study a large variety of BVDV surface motions has been documented. Interestingly, apart from a decrease in the velocity of directed motion, no effect of CD46_fluo_ on the surface transport of BVDV could be identified, indicating a minor role during this process. First insights into the progression of virus infection indicate the quick distribution of BVDV protein throughout the ER in an initial virus dose dependent manner. It will be highly interesting to identify and characterize the additional factors orchestrating BVDV entry to develop a clear picture of this process.

## Figures and Tables

**Figure 1 viruses-12-00116-f001:**
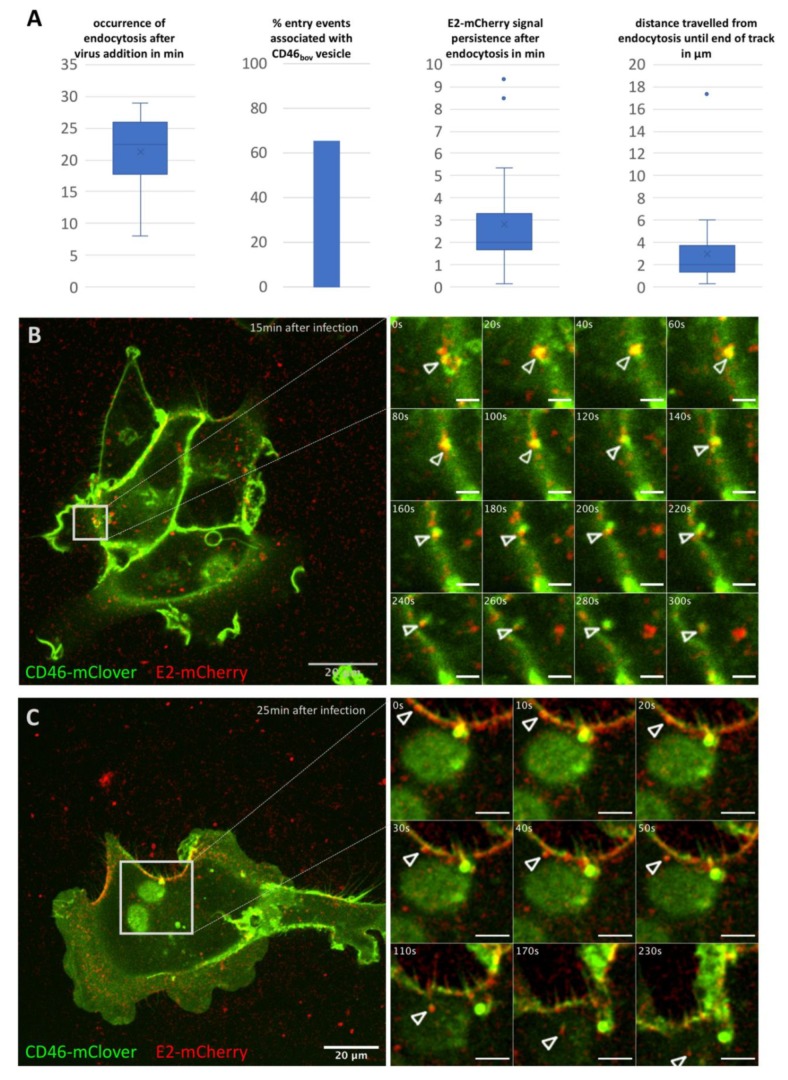
Characterization of BVDV_E2-mCherry_ entry into SK6 CD46_fluo_ cells. (**A**) Characterization of 26 entry events of BVDV_E2-mCherry_ in SK6 CD46_fluo_ cells with regard to occurrence after virus addition, association with CD46_fluo_ signal, intracellular persistence of the E2-mCherry signal and intracellular distance travelled. Outliers (Q3 + 1.5-times interquartile range) are indicated by dots. (**B**,**C**) Examples of entry events of BVDV_E2-mCherry_ (red) into SK6 CD46_bov_ (green) cells. The full field of view at the time of the start of acquisition (time after virus addition is specified in the top right corner) is shown and the area of interest is indicated by grey squares. Frames as depicted in the detail images were acquired every 10 s for up to 10 min after the start of acquisition. Times indicated in s refer to the start of acquisition. The length of the scale bar in the detail images in (**B**) = 2.5 µm and in (**C**) = 5 µm.

**Figure 2 viruses-12-00116-f002:**
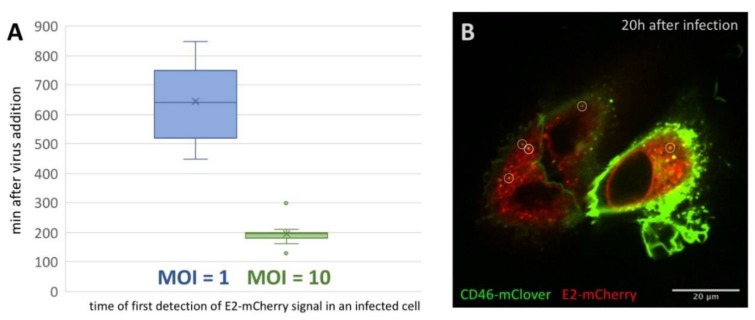
E2-mCherry signal development but not signal distribution is depending on the MOI. (**A**) Occurrence of E2-mCherry signal in min after addition of BVDV_E2-mCherry_ at an MOI of 1 (blue) or 10 (green) to SK6 CD46_fluo_ cells. SK6 CD46_fluo_ cells were imaged recording one frame every 10 min at one z-level for 16 h. The mean is indicated by x and outliers (Q3 + 1.5-times interquartile range) are indicated by dots. (**B**) Distribution of E2-mCherry (red) and CD46_fluo_ (green) signal 20 h after infection with BVDV_E2-mCherry_ in SK6 CD46_fluo_ cells. SK6 CD46_fluo_ cells were imaged for 10 min recording 3 z-levels (0.8 µm) every 10 s (Appendix A). Points of high E2-mCherry intensity that are colocalizing with CD46_fluo_ are indicated by white circles.

**Figure 3 viruses-12-00116-f003:**
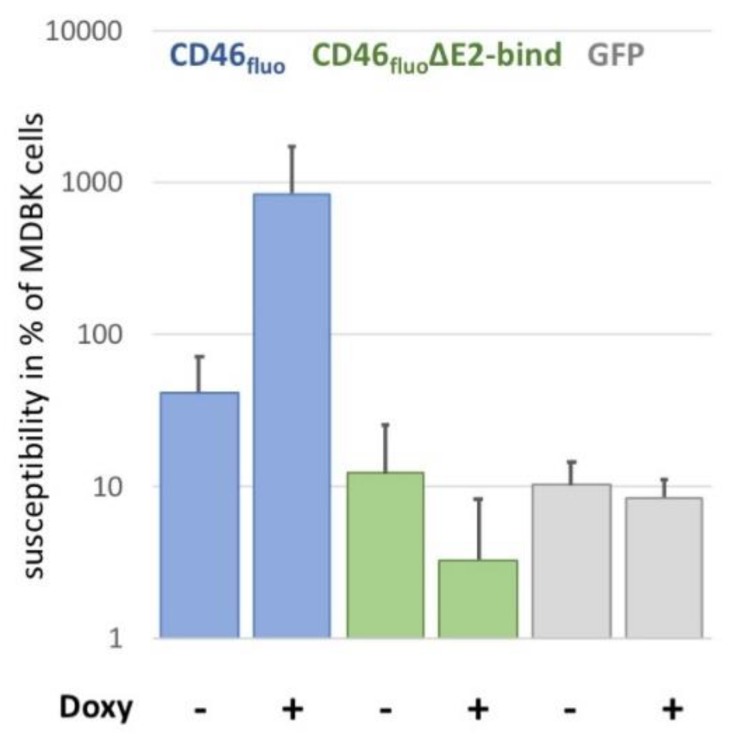
Deletion of CCP-1 reduces CD46_fluo_ effect of the susceptibility of SK6 cells to the level of GFP-expressing controls. Susceptibility of SK6 CD46_fluo_ (blue), CD46_fluo_∆E2bind (green) and GFP-expressing (grey) SK6 cells to infection with BVDV_E2-mCherry_ in % of MDBK cells (=100%) with and without the induction of expression by the addition of doxycycline (Doxy) (*n* = 6). Cells were infected with serial dilutions of BVDV_E2-mCherry_ for 4 h. Subsequently, medium was exchanged to medium containing 1% carboxymethycellulose and E2-mCherry positive foci were detected 48 h after infection by fluorescence microscopy to determine the focus forming units (ffu)/mL. Susceptibility of a given cell line was calculated as the percentage of ffu/mL determined for MDBK cells, which was set to 100%.

**Figure 4 viruses-12-00116-f004:**
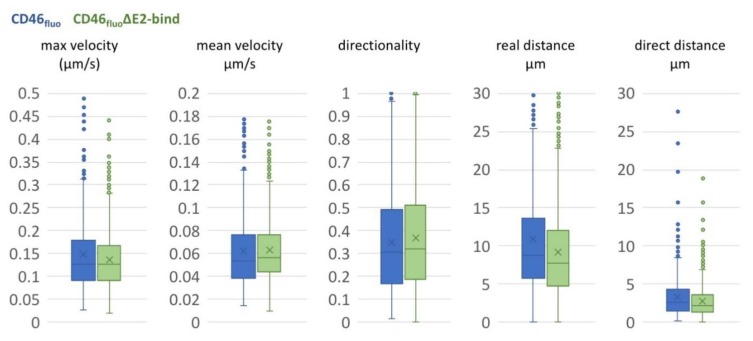
Movements of BVDV_E2-mCherry_ on the surface of SK6 CD46_fluo_ (blue) or CD46_fluo_∆E2bind cells (green) are comparable. Box and Whisker blots of maximum and mean velocity, directionality, real distance and direct distance of particles tracked on the surface of SK6 CD46_fluo_ (blue, *n* = 801) or CD46_fluo_∆E2bind cells (green, *n* = 1081), respectively. The mean is indicated by x and outliers (Q3 + 1.5-times interquartile range) are indicated by dots.

**Figure 5 viruses-12-00116-f005:**
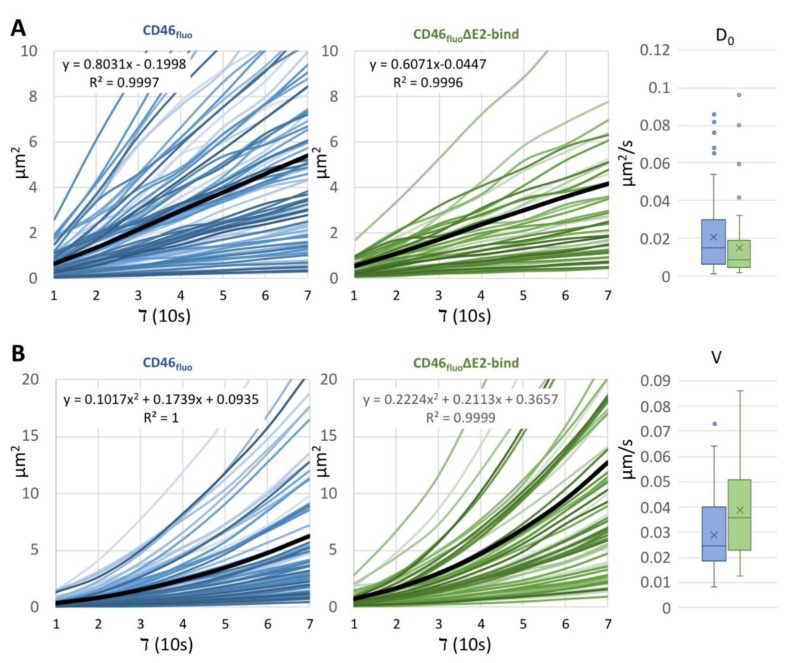
The average velocity of directed motion of BVDV_E2-mCherry_ particles is increased on the surface of SK6 CD46_fluo_∆E2bind cells. MSD analysis of particle movement on SK6 CD46_fluo_ and CD46_fluo_∆E2bind cells. (**A**) Slopes of tracks on SK6 CD46_fluo_ (blue) and CD46_fluo_∆E2bind (green) cells displaying a linear increase and distribution of the calculated diffusion coefficient D_0_ depicted as box and whisker blot. The mean is indicated by x and outliers (Q3 + 1.5 times interquartile range) are depicted as dots. Average slopes are depicted as thick black lines. (**B**) Slopes of tracks on SK6 CD46_fluo_ (blue) and CD46_fluo_∆E2bind (green) cells displaying an exponential increase and distribution of the calculated particle velocities V, depicted as box and whisker blot. The mean is indicated by x and outliers (Q3 + 1.5 times interquartile range) are depicted as dots. Average slopes are depicted as thick black lines.

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
