# Peer review of "Real Time Analysis of Bovine Viral Diarrhea Virus (BVDV) Infection and Its Dependence on Bovine CD46"

_viruses, 2020, doi:10.3390/v12010116_

Round 1
Reviewer 1 Report
The manuscript viruses-683406 „Real time analysis of BVDV infection and its dependence on CD46“ provides live cell imaging data about entry process of the pestivirus Bovine viral diarrhea virus (BVDV). It was demonstrated that interaction of BVDV with its receptor CD46 does not result in altered motion of the particle on the cell surface. The study analyses a high number of tracks to characterize the motion of BVDV particles and for the first time provides deeper insight in the cinetics of pestiviral entry. The manuscript is well written and clear in its presentation. With some minor corrections, the manuscript can be recommended for publication in viruses.
Points that should be addressed in a revised version:
Why is the study performed on a porcine cell line that is still able to express porcine CD46? Please explain. Can you really conclude that CD46 independent entry is observed as long as porcine CD46 is present on the cells, in addition to recombinant CD46∆E2bind? A bovine knockout cell line seem to be more appropriate to exclude that low number of CD46-independent entry events (2 out of 1081 tracks) are not based on using the porcine CD46. Please discuss. The entry process in general seem to be very inefficient in the applied system, if only 26 entry events could be identified in total and although CD46 is highly abundant due to over expression. How can this be explained? How would this look on a bovine cell line? Was the expression of different CD46 variants (e.g. splice variants) checked for improved infectivity? In the abstract there seem to be a contradiction: “…analysis revealed a significantly increased velocity…” and “…CD46 is not affecting BVDV cell surface motility”. This seem not fit together and is irritating. Please rewrite. In Fig 1 it is hardly possible to see any details due to small size of the photos. It might not be necessary to show photos taken every 10 seconds. Please reduce the number of photos and enlarge the remaining photos for better visualization. Abbreviations should be introduced when mentioned first time, e.g. MSD analysis, MLV, TIM-1, MOI, SNR and others. Please check whole manuscript. Please always insert a space between value and dimension(e.g. please change 0.062µm into 0.062 µm, 10sec in 10 sec, etc.)
Author Response
Comments to the reviewers:
We sincerely thank both reviewers for their time and critical feedback regarding this manuscript and hope that we were able to improve the manuscript based on their comments.
Reviewers’ comments are shown in black, our answers in blue and changes made to the manuscript text are in italics.
Reviewer 1:
Why is the study performed on a porcine cell line that is still able to express porcine CD46? Please explain. Can you really conclude that CD46 independent entry is observed as long as porcine CD46 is present on the cells, in addition to recombinant CD46∆E2bind? A bovine knockout cell line seem to be more appropriate to exclude that low number of CD46-independent entry events (2 out of 1081 tracks) are not based on using the porcine CD46. Please discuss.
We thank the reviewer for raising these questions. To our knowledge, porcine CD46 is not involved in BVDV invasion (Krey 2006a) and therefore the utilisation of a porcine cell line to examine CD46 properties regarding BVDV entry is an accepted practice. We agree that a knock-out cell line would be the neatest test system, but this test system is currently not available to us.
Bovine CD46 though seems not to be the only mean for BVDV to enter its host cells. Cells of other cloven-hoofed animals can also be infected, but with much lower efficacy and anti-CD46 sera are not able to completely block BVDV entry (Maurer 2004, Krey 2006a). Therefore, the low rate of entry events observed might be due to the utilisation of an alternative, yet unknown receptor with a lower uptake efficacy or an alternative entry pathway. This is also supported by the recent discovery that BVDV cell-to-cell spread is independent of CD46 (Merwaiss 2019).
To clarify this, the results section has been modified as follows (lines 173-177):
SK6 cells inducibly expressing the fluorophore labelled, cellular surface receptor, bovine CD46 were chosen as system to study the infection cycle, as these cells demonstrated a high susceptibility to infection with BVDV after induction. SK6 cells not expressing bovine CD46 display a low susceptibility to BVDV infection, which is not mediated by porcine CD46, as porcine CD46 has previously been shown not to be involved in BVDV invasion[16].
The entry process in general seem to be very inefficient in the applied system, if only 26 entry events could be identified in total and although CD46 is highly abundant due to over expression. How can this be explained? How would this look on a bovine cell line? Was the expression of different CD46 variants (e.g. splice variants) checked for improved infectivity?
We were also startled by the low number of entry events we could observe. One possibility is that the preferred sites of virus entry are located on the cell body, in areas that were not probed in our experimental set-up, as we were mostly interested in elucidating particle transport from the cell periphery to the cell body. Particles could frequently be observed leaving the area of acquisition towards parts located higher on the cell body. Therefore, we likely missed entry events as more focal planes could unfortunately not be sampled due to phototoxicity and short frame times. Additionally, the specific infectivity of BVDV particles is rather low and it is possible that particles present with defects that allow for binding and transport, but not for uptake.
We did not check the effect of different splice variants or allelic versions of CCP1 on particle transport due to the sheer amount of data needing analysis. The splice version utilized in this experimental setup encodes for the full length C-terminus and represents the same splice variant as reported by Maurer et al 2004.
The following paragraph was added to the discussion to address the low number of entry events and the utilisation of bovine cells (lines 370-383):
In general, the number of observable entry events into SK6 CD46fluo was very low (2.8% of tracked particles). This might indicate that cellular uptake of BVDV is a rather inefficient process. However, during data acquisition, we were not able to sample the whole cellular surface due to the short frame interval and phototoxicity. The focus of acquisition was on the cellular surface in contact with the growth support, to clearly depict cellular protrusions, the lamellipodium and the lamella. Particles could frequently be observed exiting the acquisition planes to regions of the cell body wall that were inaccessible in the chosen acquisition scheme. It is therefore possible that the low amount of observed entry events is – at least partially – due to a preference of BVDV to enter cells at specific areas of the cell body wall. Also, the specific infectivity of BVDVE2-mCherry is below 1:10, meaning that more than 10 genome equivalents are needed for a successful establishment of infection. It is currently unclear what particle defects are responsible for this phenomenon, but it has to be taken into consideration that a certain number of particles are already unable to evoke cellular uptake. In order to clarify these open questions and to also examine the entry process of BVDV into primary host cells, MDBK CD46 knock-out cell lines would be a valuable tool for future elucidation of the entry process.
The following paragraph was added to the discussion to address the allelic and splice variants of bovine CD46 (lines 391-400):
The importance of factors different from CD46 in BVDV propagation is also highlighted by recent results reporting that BVDV cell-to-cell spread is independent of CD46[12]. The CD46 splice variant employed in this study is identical to the bovine CD46 variant initially reported by [15]. Unfortunately, we were not able to elucidate the effects of different allelic versions of CCP-1 and different CD46 splice variants on the dynamics of BVDV entry due to the huge efforts of data analysis. A different permissiveness to infection of these different allelic and splice variants has however already been demonstrated [30] and their effect on the dynamics of BVDV entry will be an interesting topic of future studies.
In the abstract there seem to be a contradiction: “…analysis revealed a significantly increased velocity…” and “…CD46 is not affecting BVDV cell surface motility”. This seem not fit together and is irritating. Please rewrite.
We thank the reviewer for this statement. Lines 25-29 of the abstract have been rewritten as follows to mitigate this contradiction:
These results indicate that the presence of bovine CD46 is only affecting the speed of directed transport, but otherwise not influencing BVDV cell surface motility. Instead, bovine CD46 seems to be an important factor during uptake, suggesting the presence of additional cellular proteins interacting with the virus which are able to support its transport on the virus surface.
In Fig 1 it is hardly possible to see any details due to small size of the photos. It might not be necessary to show photos taken every 10 seconds. Please reduce the number of photos and enlarge the remaining photos for better visualization.
We have reduced the number of images and increased their size for each series.
Abbreviations should be introduced when mentioned first time, e.g. MSD analysis, MLV, TIM-1, MOI, SNR and others. Please check whole manuscript. Please always insert a space between value and dimension (e.g. please change 0.062µm into 0.062 µm, 10sec in 10 sec, etc.)
The manuscript has been rewritten accordingly and we sincerely hope that all raised points have been corrected appropriately.
Reviewer 2 Report
The authors describe a detailed analysis of the entry process used by the BVDV virus by employing fluorescent tagged virus and CD46 receptor expression in cells (SK6) that are normally not permissive to BVDV infection.
Although the authors amassed a huge quantity of images and did a comprehensive analysis of the data collected, it is not clear to me why the authors did so? Hence, it would advise the authors to consider the following suggestions to further improve the manuscript. In addition I attached a pdf file in which I annotated additional comments.
Suggestions for major improvements:
The introduction should be modified to include paragraphs on CD46 as a receptor for pathogens and its structure. The unknowns in the BVDV entry process. The authors should also indicate what the aim of their research is; the main research question they aimed to answer. Now, the introduction ends with a paragraph the describes the method the researchers apply; without details about why they are doing this research or which answers they are after. In the method section the authors should disclose the details about the molecular infectious BVDV cDNA clone they use and how they prepare virus from this clone. Also, they should compare their clone with others used to for example elucidate cell to cell spread of BVDV. In the results sections the authors should use conclusions of different experiments as titles of paragraphs and captions of figures. As such it becomes much more clear to the reader what the authors findings are. Especially, for the figures this is essential given that they should be interpretable independent of the text. The concluding paragraph of the discussion should be added or at least the authors should improve the last paragraph of the discussion to summarize what there findings were and how the contributed to the field of BVDV entry.
Author Response
Comments to the reviewers:
We sincerely thank both reviewers for their time and critical feedback regarding this manuscript and hope that we were able to improve the manuscript based on their comments.
Reviewers’ comments are shown in black, our answers in blue and changes made to the manuscript text are in italics.
Reviewer 2:
Suggestions for major improvements:
The introduction should be modified to include paragraphs on CD46 as a receptor for pathogens and its structure.
We thank the reviewer for his suggestion. The following paragraph has been added to address the reviewer’s request.
Line 74-84:
Bovine CD46 is a ubiquitously expressed, type I transmembrane glycoprotein and exists as different splice variants, affecting the length of its heavily O-glycosylated, membrane poximal regions (STP) and its cytoplasmic C-terminus. The membrane distal, extracellular part of the protein consists of 4 complement control protein modules (CCP) which have been implicated in the binding of a variety of pathogens, leading to its description as a “pathogens magnet”[20]. Physiologically, CD46 is a cofactor of the inactivation of complement components C3b and C4b and also involved in T cell regulation, modulation of autophagy and reproductive biology (reviewed in [21]). Its surface levels are regulated by clathrin-dependent endocytosis[21] or micropinocytosis after cross-linking[22]. BVDV E2 binding to CD46 is mediated by the 30 C-terminal amino acids of CCP-1[16]. Interestingly, CD46’s physiological ligands mainly interact with the membrane proximal CCPs 3-4, whilst pathogens mostly interact with the membrane distal CCPs 1-2 [23].
The unknowns in the BVDV entry process.
The authors should also indicate what the aim of their research is; the main research question they aimed to answer. Now, the introduction ends with a paragraph the describes the method the researchers apply; without details about why they are doing this research or which answers they are after.
We thank the reviewer for raising these points. The passage was modified as follows:
Line 85-88:
Whilst the function of bovine CD46 as a receptor for BVDV is well documented in the literature, it is currently unclear what its exact functions are during virus attachment and entry. Also, it is not sufficient for entry. To determine which stages of the entry process – attachment, surface motion or uptake - are affected by CD46, we conducted life cell imaging experiments employing purified, fluorophore-E2 labelled BVDV particles and SK6 cell lines inducibly expressing fluorophore labelled CD46 with and without the virus interacting CCP-1 module[16].
In the method section the authors should disclose the details about the molecular infectious BVDV cDNA clone they use and how they prepare virus from this clone. Also, they should compare their clone with others used to for example elucidate cell to cell spread of BVDV.
We thank this reviewer for his suggestions to clarify the methodology. The following passages were added to address them:
Line 70-73:
The BVDV-1 clone employed in [12] also encodes for an mCherry-E2 fusion protein, in the backbone of strain NADL, demonstrating the utilization of different transmission modes in the presence of an E2-fusion protein.
Line 125-126:
The sequence of the plasmid encoding the full sequence of BVDVE2-mCherry is provided as supplementary data.
In the results sections the authors should use conclusions of different experiments as titles of paragraphs and captions of figures. As such it becomes much more clear to the reader what the authors findings are. Especially, for the figures this is essential given that they should be interpretable independent of the text.
We thank this reviewer for his guidance. As highlighted in the pdf kindly provided by this reviewer, the figure captions have been adapted as follows:
Fig.1: Characterisation of BVDVE2-mCherry entry into SK6 CD46fluo cells.
Fig.2: E2-mCherry signal development but not signal distribution is depending on the MOI.
Fig.3: Deletion of CCP-1 reduces CD46fluo effect on the susceptibility of SK6 cells to the level of GFP-expressing controls.
Fig.4: Movements of BVDVE2-mCherry on the surface of SK6 CD46fluo (blue) or CD46fluo∆E2bind cells (green) are comparable.
Fig.5: The average velocity of directed motion of BVDVE2-mCherry particles is increased on the surface of SK6 CD46fluo∆E2bind cells.
The subheadings of the results sections have been modified as follows:
3.1 BVDV entry is a slow process
3.2 First detection of BVDV E2-mCherry signal is depending on MOI
3.3 CD46fluo decreases the speed of directed surface motion of BVDVE2-mCherry
The concluding paragraph of the discussion should be added or at least the authors should improve the last paragraph of the discussion to summarize what there findings were and how the contributed to the field of BVDV entry.
We thank the reviewer for this remark and hope that the following modification of the last paragraph of the discussion sufficiently addresses this (lines 421-427):
In the presented study a large variety of BVDV surface motions has been documented. Interestingly, apart from a decrease in the velocity of directed motion, no effect of CD46fluo on the surface transport of BVDV could be identified, indicating a minor role during this process. First insights into the progression of virus infection indicate the quick distribution of BVDV protein throughout the ER in an initial virus dose dependent manner. It will be highly interesting to identify and characterize the additional factors orchestrating BVDV entry to develop a clear picture of this process.
Suggestions in the attached pdf file:
The abbreviation BVDV has been replaced in the title by the full virus name. However, we were unfortunately not able to replace all abbreviations in the abstract, as its length is already exceeding the suggested maximum word count.
Reviewer 3 Report
The manuscript with title ''Real time analysis of BVDV infection and its dependence on CD46'' is well written and suitable for publication. I have only two minor remarks: 1. Abstract-explain the meaning of MSD analysis 2. Through manuscript there are several mistakes (space missing), eg. Line 45, line 46, line 52, line 69, line 84...Author Response
We are very grateful for the positive feedback of this reviewer regarding our manuscript and would like to thank him/her for his/her time and effort.
The spelling mistakes have been corrected and all abbreviations have been introduced.